# Dehydroascorbic Acid Affects the Stability of Catechins by Forming Conjunctions

**DOI:** 10.3390/molecules25184076

**Published:** 2020-09-07

**Authors:** Lin Chen, Wei Wang, Jianyong Zhang, Weiwei Wang, Dejiang Ni, Heyuan Jiang

**Affiliations:** 1Key Laboratory of Tea Biology and Resources Utilization, Ministry of Agriculture, Tea Research Institute, Chinese Academy of Agricultural Sciences, 9 Meiling South Road, Hangzhou 310008, China; chl911003@tricaas.com (L.C.); ww1040491839@163.com (W.W.); zjy5128@tricaas.com (J.Z.); wangwei11211@tricaas.com (W.W.); 2Department of Tea Science, College of Horticulture and Forestry Science, Huazhong Agricultural University, Wuhan 430070, China

**Keywords:** catechins, ascorbic acid, dehydroascorbic acid, stability, mechanism

## Abstract

Although tea catechins in green tea and green tea beverages must be stable to deliver good sensory quality and healthy benefits, they are always unstable during processing and storage. Ascorbic acid (AA) is often used to protect catechins in green tea beverages, and AA is easily oxidized to form dehydroascorbic acid (DHAA). However, the function of DHAA on the stability of catechins is not clear. The objective of this study was to determine the effects of DHAA on the stability of catechins and clarify the mechanism of effects by conducting a series of experiments that incubate DHAA with epigallocatechin gallate (EGCG) or catechins. Results showed that DHAA had a dual function on EGCG stability, protecting its stability by inhibiting hydrolysis and promoting EGCG consumption by forming ascorbyl adducts. DHAA also reacted with (−)-epicatechin (EC), (−)-epicatechin gallate (ECG), and (−)-epigallocatechin (EGC) to form ascorbyl adducts, which destabilized them. After 9 h of reaction with DHAA, the depletion rates of EGCG, ECG, EC, and EGC were 30.08%, 22.78%, 21.45%, and 13.55%, respectively. The ability of DHAA to promote catechins depletion went from high to low: EGCG, ECG, EGC, and EC. The results are important for the processing and storage of tea and tea beverages, as well as the general exploration of synergistic functions of AA and catechins.

## 1. Introduction

Tea catechins in green tea and green tea beverages make important contributions to flavor quality and health functions. Catechins belong to the group of flavanols and account for about 70% of the total amount of polyphenols [1,2]. There are four major catechins in tea leaves and green tea: (−)-epicatechin (EC), (−)-epicatechin gallate (ECG), (−)-epigallocatechin (EGC), and (−)-epigallocatechin gallate (EGCG) [3]. Generally, EGCG accounts for 50%–80% of the total catechins, exhibiting a number of beneficial properties for human health, including antioxidant and anticarcinogenic activity [4,5], as well as anti-inflammatory [6,7] and antidiabetic properties [8]. Therefore, the stability of tea catechins has attracted increased attention in recent years, especially EGCG.

The addition of ascorbic acid (AA) to green tea beverages is often used to protect catechins’ stability. AA has both antioxidative properties and a broad range of bioactivities that protect against endogenous oxidative DNA damage [9]; reduce the risk of schizophrenia, major depressive disorder, and bipolar disorder [10,11]; and lower central blood pressure [12]. AA can both protect the stability of catechins and synergize with catechins to enhance health functions. Xie found that when supplemented with tea polyphenols, AA can improve blood uric acid, lipid metabolism, and oxidative stress in hyperuricemia and dyslipidemia patients [13]. Some studies have indicated that the inhibitive effect on Helicobacter pylori of tea polyphenol mixed with AA is more efficient than that of separate individuals [14]. AA has antioxidant properties and can be rapidly oxidized in aqueous solutions to form dehydroascorbic acid (DHAA). Researchers have reported that the conversion pathway of AA in an aqueous solution, AA, is oxidized to form DHAA [15], and DHAA is hydrolyzed to 2,3-diketogulonic acid (DKG). DKG can be further oxidized to over 50 substances [16]. Therefore, elucidating the effect of DHAA on the stability of catechins is important for a comprehensive understanding of AA and catechin interaction.

Some research found that AA significantly increased the stability of green tea catechins (GTC) [17]. Su found that the addition of AA destabilized GTC and theaflavins (TFs) compared with a control group in a 6-month stability assessment of GTC and TFs. GTC almost disappeared after the first month [18]. Hung detected ascorbyl adducts of EGCG, 6C-DHAA-EGCG, and 8C-DHAA-EGCG, in both green tea and oolong tea beverages [17]. Zhu [19] found that tea catechins could block the advanced glycation of lens crystallins induced by dehydroascorbic acid by generating ascorbyl adducts of catechins. AA was used to prevent anthocyanins color loss and improve the perceived nutritional quality of beverages. However, it was reported that AA could also destabilize anthocyanins, even if the destruction mechanism is not clear. In any case, the prevailing hypothesis is that the AA oxidation process could cleave the structure of anthocyanins [20]. It can be assumed that DHAA may destabilize the stability of catechins by generating ascorbyl adducts with catechins.

A series of experiments using AA and DHAA with EGCG or catechins was conducted to both clarify the effect of DHAA on catechin stability and analyze the mechanism.

## 2. Results

### 2.1. EGCG Stability at 25 °C

After storing for 20 days, the substrates and products in the incubation system were analyzed using ultra-performance liquid chromatography coupled with a photo-diode array detector and QDA mass detector (UPLC-PDA-QDA); their representative chromatograms are shown in Figure 1A. Classical negative ESI (Electrospray Ionization) mass spectra of gallic acid (GA), EGC, and EGCG are also shown in Figure 1A to verify the reliability of the qualitative analysis. Moreover, the time course of EGCG and products is shown in Figure 1B. As shown in Figure 1B, gallocatechin gallate (GCG), ECG, EGC, and GA were detected after 5 days. EGC, ECG, and GA are hydrolysates of EGCG, and GCG is isomer of EGCG. EGCG underwent hydrolysis and epimerization during storage at 25 °C. The concentration of EGCG decreased, while EGC and GA increased and then decreased during storage.

The EGCG degradation was assumed to follow first-order kinetics. The degradation kinetics of EGCG were expressed by first-order kinetics: Ct/C0 = exp(−kt),(1)
where Ct and C0 are the concentration of EGCG at time t and time = 0, respectively. As shown in Figure 1C, the rate constant of first-order kinetics was calculated to best fit the experiment results. It can be seen that the R^2^ value was 0.979, which indicated the validity of the assumption that EGCG degradation followed the first-order kinetics during storage.

To clarify the EGCG stability during storage, the changes of EGCG were divided into two parts: the reserve part and the degradation part. The degradation part of EGCG can also be summarized by three pathways, i.e., epimerization, hydrolysis, and oxidation. During our research, EGC, ECG, and GA were only produced by the hydrolysis of EGCG. Therefore, the percentage of hydrolysis was calculated by the concentration of EGC, ECG, and GA. 

The percentage of EGCG epimerization was calculated by the GCG concentration. Little dimers, TFs, and theasinensins (TSs) were detected in the experiments, but they were unstable and quickly disappeared. The color of the incubation system changed from colorless to brown. It meant that EGCG underwent oxidation during storage. In this research, the percentage of the oxidation pathway was counted by removing reserve, epimerization, and hydrolysis pathway percentage.

The percentage of the different EGCG conversion pathways at 20 and 30 days were shown in Figure 1C. The percentage of reserve, epimerization, hydrolysis, and oxidation at 20 days were 20.46%, 0.83%, 40.48%, and 38.23%, respectively. Results showed that hydrolysis and oxidation were the main pathways for EGCG degradation, with hydrolysis accounting for the majority. The percentage of the reserve, epimerization, hydrolysis, and oxidation at 30 days were 1.63%, 0%, 36.16%, and 62.21%, respectively. Hydrolysis and oxidation were also the main pathways of degradation. Yet the percentage of hydrolysis decreased, and the percentage of oxidation increased. Hydrolysates produced by EGCG hydrolysis may oxidize with time. It was speculated that DHAA could significantly affect EGCG stability by influencing the EGCG hydrolysis and oxidation pathways.

### 2.2. Effect of Different Concentration of DHAA on EGCG Stability

EGCG incubation systems with different DHAA concentrations were stored at 25 °C to analyze the effect of DHAA on EGCG stability. Figure 2A showed apparent first-order kinetics of EGCG degradation with different DHAA concentration treatments. The rate constant of EGCG degradation with 20 µg/mL and 200 µg/mL of DHAA were 0.0581 and 0.0605, which were lower than CK (control group without DHAA). The rate constant of EGCG degradation with 2000 µg/mL DHAA was 0.137, which was higher than that of CK. This meant that 20 µg/mL and 200 µg/mL of DHAA protected EGCG stability and 2000 µg/mL of DHAA accelerated the EGCG degradation. Therefore, the effect of DHAA on the hydrolysis and oxidation products of EGCG was analyzed to explain the result.

The effect of different DHAA contents on the concentration of GA at 30 days is shown in Figure 2B. It can be seen that DHAA significantly reduced GA production compared with the control group (*p* < 0.05). This implies that DHAA inhibited EGCG hydrolysis to protect its stability. The higher the DHAA concentration, the stronger the inhibitory effect of hydrolysis.

Figure 2C showed the changes of DHAA–EGCG adducts areas of three DHAA treatments. DHAA–EGCG adducts were ascorbyl conjugates of EGCG, including 8C–DHAA–EGCG and 6C–DHAA–EGCG, which can be detected in green tea and oolong tea beverages [17]. It can be seen that DHAA–EGCG adducts were detected in all three treatments. DHAA had the potential to destabilize EGCG stability by forming DHAA–EGCG adducts. The highest peak area of DHAA–EGCG under 2000 µg/mL DHAA treatment was about 180,000, which was nearly 10 times higher than that under 200 µg/mL DHAA treatment. DHAA–EGCG production was positively correlated with the DHAA concentration.

The effect of different DHAA contents on the highest concentration of GCG during storage is shown in Figure 2D. There was a significant difference in GCG content between four treatments (*p* < 0.05). However, the maximum GCG content did not exceed 5 µmol/L, and the epimerization pathway had little effect on EGCG stability.

Hydrolysis and oxidation were the main degradation pathway when EGCG was stored at 25 °C. The inhibitory effect of 20 µg/mL and 200 µg/mL DHAA on the hydrolysis of EGCG was stronger than the effect of the promoting oxidation, resulting in the phenomenon that DHAA could protect EGCG stability. While the promoting effect of 2000 µg/mL DHAA on the oxidation of EGCG was stronger than the inhibitory effect on hydrolysis, 2000 µg/mL DHAA destabilized the EGCG stability. These results validated the previous hypothesis.

### 2.3. Effect of AA and DHAA on the EGCG Stability

A series of systems investigated the mechanism of forming DHAA–EGCG, including AA and EGCG incubation, DHAA and EGCG incubation, and AA, DHAA, and EGCG incubation. The AA and DHAA concentration were 2000 µg/mL. Figure 3A shows the first-order kinetics for the EGCG degradation of four incubation systems. The k value for EGCG degradation with 2000 µg/mL of AA was 0.058, which was lower than that of CK. Therefore, 2000 µg/mL of AA protected the EGCG stability. It was speculated that the AA antioxidative activity inhibited EGCG oxidation. The k value for EGCG degradation with 2000 µg/mL of AA and 2000 µg/mL of DHAA was 0.137, which was higher than CK but lower than the treatment with 2000 µg/mL of AA. Figure 3B shows that the percentage of oxidation pathway with AA and DHAA treatment was significantly lower than that with the DHAA treatment. Results showed that AA and DHAA, when coexisting, played a protective role in the EGCG stability. However, the protective effect was weaker than AA alone. AA and DHAA, when coexisting, also promoted the oxidation of EGCG, but the effect was lower than DHAA itself.

The chemical pathway for the production of catechins ascorbyl conjugates was reported to successively undergo two-step nucleophilic additions at active sites [19]. As shown in Figure 4, the deprotonation of OH-5 or OH-7 in the EGCG A ring generated a carbanion species at C-6 or C-8 as a nucleophile, which attacked the carbonyl group at C-2 in DHAA to generate the key intermediate 6C- or 8C-adduct, respectively. Next, nucleophilic OH-5 or OH-7 in the EGCG A ring attacked the carbonyl group at C-3 in the DHAA residue to close the furan ring, thereby producing 6C–DHAA–EGCG or 8C–DHAA–EGCG. It was speculated that AA could inhibit the first step of the reaction, resulting in the phenomenon that AA and DHAA could protect EGCG when coexisting.

### 2.4. Formation of Ascorbyl Adducts of Four Catechins

The formation of ascorbyl adducts of catechins (EGCG, EGC, ECG, and EC) was evaluated in citrate phosphate buffer (pH 5.6) at 37 °C. New products were rapidly detected in all incubation systems, and the liquid chromatograph profile and mass spectrum of products are shown in Figure 5. The *m/z* of precursor and fragment ion of the substrates and products is shown in Table 1. Their structures were established by analyzing the negative mass spectra in combination with what was reported in the literature [19].

As shown in Figure 5A(2), the molecular weight of DHAA–EGCG, synthesized from EGCG and DHAA, was 632.2 mass units based on negative mass spectrum at *m*/*z* 631.2 [M – H], which was 174 (mass weight of DHAA, 175) higher than EGCG. The *m*/*z* of fragment ion was 513.1, 495.2, and 343.1. The mass spectral features of DHAA–EGCG reported in our study were similar with previous research. Hung [17] found that the MS spectra of purified 8C–DHAA–EGCG had 513.1, 495.1, and 343.1 *m*/*z*. Zhu [19] reported that the major product ions of 8C–DHAA–EGCG were 512.7 and 494.8 *m*/*z*. All mass spectrum in this study supported new products such as ascorbyl conjunctions, i.e., 6C–DHAA–EGCG and 8C–DHAA–EGCG.

As shown in Figure 5B(2), the molecular weight of DHAA–ECG, synthesized from ECG and DHAA, was 616.2 mass units based on the negative mass spectrum at 615.2 *m*/*z* [M − H], which was 174 higher than ECG. The *m*/*z* of a fragment ion was 479.1 and 327.1. The *m*/*z* of a DHAA–ECG fragment ion was 16 units less than DHAA–EGCG. The molecular weight of EGCG was 16 (mass weight of –OH, 17) mass units higher than ECG. Therefore, new products were identified as ascorbyl conjunctions of ECG, including 6C–DHAA–ECG and 8C–DHAA–ECG.

As shown in Figure 5C(2), the molecular weight of DHAA–EC, synthesized from EC and DHAA, was 464.2 mass units based on the negative mass spectrum at 463.2 *m*/*z* [M − H], which was 174 units higher than EC. The molecular weight of DHAA–EC was 152 units less than DHAA–ECG, which corresponded to the loss of the gallate group. The *m*/*z* of a fragment ion was 327.1 and 191.0. The DHAA–EC fragment ion was 327.1 *m*/*z* and 152 units less than DHAA–ECG (479.1), which corresponded to the loss of the gallate group. Therefore, the new products were identified as ascorbyl conjunctions of EC, including 6C-DHAA-EC and 8C-DHAA-EC.

As shown in Figure 5D(2), the molecular weight of DHAA-EGC, synthesized from EGC and DHAA, was 480.1 mass units based on the negative mass spectrum at *m*/*z* 479.1 [M − H], which was 174 units higher than EGC. The *m*/*z* of a fragment ion was 451.2 and 343.1. Zhu [19] reported that the ascorbyl conjunctions of EGC had a characteristic fragment ion at 343.0 *m*/*z*, which was based on negative ESI-MS. Therefore, the new products were identified as EGC ascorbyl conjunctions, i.e., 6C–DHAA–EGC and 8C–DHAA–EGC.

The consumption rate of four catechins is shown in Figure 5E,F. Consumption rate demonstrated the ability of DHAA to capture different catechins. Results showed that the rate of catechins consumption gradually increased with time and then became stable. The consumption rates of EGCG, EGC, EC, and ECG were 30.08%, 13.55%, 21.45%, and 22.78% after incubation for 540 min, respectively. Among four catechins, the ability of DHAA to capture catechins was, from high to low: EGCG, ECG, EC, and EGC. The ability of DHAA to trap catechins indicated that DHAA could promote the degradation of catechins of tea or tea beverages when stored. The difference in the ability of DHAA trapping catechins induced the difference in destabilizing the catechin effect.

## 3. Discussion

The results showed that DHAA had dual effects on EGCG stability, protection, and destruction. DHAA can protect the EGCG stability by inhibiting the hydrolysis, while also promoting the consumption by forming DHAA–EGCG adducts. When AA and DHAA coexist, AA could inhibit the formation of DHAA–EGCG by inhibiting the first step. DHAA could trap EGCG, ECG, EC, and EGC to form ascorbyl adducts of catechins. Among the four catechins, the ability of DHAA to capture catechins was, from high to low: EGCG, ECG, EC, and EGC. DHAA could promote the degradation of catechins of tea, tea beverages, and tea foods by forming ascorbyl adducts. 

Zhu [19] conducted experiments in which flavanols prevented a DHAA-induced advanced glycosylation of lens protein. The results showed that EGCG exerted the best DHAA-trapping activity, followed by EGC, ECG, and EC. This was inconsistent with our results. In Zhu’s research, the reaction substrates were dissolved in a buffer solution (pH 8.0) and then incubated at 37 °C. Higher pH may be the reason for the difference in the results of the two groups of experiments. In this study, the ability of DHAA to trap catechins was evaluated under the simulated tea soup or tea beverage pH conditions (pH 5–6). The ability of DHAA to capture catechins was from high to low: EGCG, ECG, EC, and EGC. It was speculated that the gallate group of the C ring affected the reactive activity between DHAA and catechins. The mechanism of the structure of catechins that affected DHAA capturing ability was worth studying in the future. New products were detected in the incubation systems of DHAA with EC and EGC. Other products could be formed by different reaction processes between DHAA and EC or EGC.

The mechanism of DHAA inhibiting hydrolysis of EGCG was not clear. It may be related to the combination of DHAA and EGCG. The reaction of the carbonyl group at C-2 in DHAA to generate the key intermediate 6C- or 8C-adduct inhibited the loss of the gallate group.

Liu [21] analyzed the contents of tea polyphenols (TP) and AA in four kinds of tea beverages without opening during storage at room temperature. The results showed that TP contents decreased significantly at the third month and AA decreased sharply in 1–3 months. It was speculated that DHAA, formed by the oxidation of AA in 1–3 months, promoted the significant degradation of TP after the third month. From this literature, it is known that the degradation rate of AA is higher than that of catechins when tea beverages are stored at room temperature and sealed, so the shelf life of tea beverages should consider the consumption of AA in combination with the results DHAA destabilized catechins.

Su [18] found that addition of AA destabilized GTC and TFs compared with the control group in a 6-month assessment of GTC and TFs stability. GTC almost disappeared after the first month. Podmore, Herbert, and Zhu found that AA could protect at the first month and then promote the degradation of Longjing GTC (1996, 1998, and 2003). This phenomenon can be explained by the different functions of AA and DHAA on EGCG stability proposed in our research. AA protected catechins stability by inhibiting oxidation. AA was oxidized to DHAA, and then DHAA promoted the consumption of catechins by forming ascorbyl catechins during storage.

In recent years, the use of green tea extracts in foods such as bread, cereals, cakes, biscuits, dairy products, noodles, instant noodles, confectionery, and ice cream improved the marketing potential for these foods [22]. These foods often contain ascorbic acid or require the addition of ascorbic acid for quality or functional improvement. The result that DHAA could trap catechins to form ascorbyl conjunctions is helpful for exploring the interaction between AA and catechins during food processing.

Research reported that AA can cause strand breakage in DNA in the presence of oxygen and AA can increase the OH radical in Fenton systems [23]. Many of the phenomenon may actually be caused by DHAA properties.

In conclusion, DHAA showed the properties of destabilizing catechins. Therefore, when AA was used to protect the stability of catechins in tea beverages and tea foods or synergize with catechins to improve health functions, the oxidation of AA and the properties of DHAA must be taken seriously.

## 4. Materials and Methods 

### 4.1. Chemicals and Reagents

DHAA was purchased from Sigma (Louis, MO, USA) and AA was purchased from Aladdin (Shanghai, China). Purified EGCG powder (PEP) was purchased from Sigma, containing 98% EGCG. Purified EGC, ECG, and EC powders were purchased from Shanghai Yuanye biological technology company (Shanghai, China). UPLC grade of methanoic acid and acetonitrile were purchased from Merck (Darmstadt, Germany). An ACQUITY UPLC BEH C18 (Waters, MA, USA) column was used in chromatographic. Citric acid and disodium hydrogen phosphate were purchased from Sigma. Detail information of abbreviations and full name of regents is provided in Appendix A.

### 4.2. Experimental Treatment and Sample Preparation

Sharma [22] found that tea catechins were relatively stable in acidic solution. Xu [24] found that the pH of extracted green tea infusion was 5.53, and Mendel [25] reported that in order to reduce the degradation of catechins in tea drinks, the pH was usually controlled at 5.5. Thus, citric acid–phosphate buffer at pH 5.6 was chosen for the stability study.

#### 4.2.1. Effects of Dehydroascorbic Acid Concentration on the EGCG Stability

Citric acid–phosphate buffer and EGCG solution was prepared as described above. In total, 3 mg, 30 mg, or 300 mg of DHAA was dissolved in 5 mL of a citric acid–phosphate buffer. Then, DHAA solutions were mixed with the EGCG solution and citric acid–phosphate buffer at the breaker. Four incubation systems were prepared; the total volume was 150 mL. The concentration of EGCG was 229 µg/mL, and the concentration of DHAA was 0 µg/mL, 20 µg/mL, 200 µg/mL, or 2000 µg/mL, respectively. The 150 mL mixed solution was divided into three parts and stored in a 50-mL centrifuge tube. The centrifuge tube was stored in incubator at 25 °C for 0, 5, 15, 20, 25, and 30 days, after which 2 mL of the mixture was taken and then filtered through a 0.22-µm filter for UPLC detection.

#### 4.2.2. Effect of Ascorbic Acid and Dehydroascorbic Acid on the EGCG Stability

The citric acid–phosphate buffer and EGCG solution was prepared as described above. In total, 600 mg of AA and 600 mg of DHAA were respectively dissolved in 10 mL of a citric acid–phosphate buffer. Four incubation systems were prepared; the total volume was 150 mL. The concentration of EGCG was 229 µg/mL. One of the reaction systems was used as the control without adding any AA or DHAA. Two of the reaction systems were respectively added with AA and DHAA, and the concentrations were all 2000 µg/mL. In the last incubation system, AA and DHAA were added simultaneously, and the concentrations were 2000 µg/mL. The 150 mL mixed solution was divided into three parts and stored in a 50-mL centrifuge tube. The centrifuge tube was stored in incubator at 25 °C for 0, 5, 15, 20, 25, and 30 days, after which 2 mL of the mixture was taken and then filtered through a 0.22-µm filter for UPLC detection.

#### 4.2.3. Formation of Ascorbyl Adducts of Four Catechins

The citric acid–phosphate buffer was prepared as described above. EGCG, EGC, ECG, and EC were dissolved in 10 mL of a citric acid–phosphate buffer and stored at 4 °C in a refrigerator, determining the concentration by UPLC. In total, 1200 mg of DHAA was dissolved in 20 mL of a citric acid–phosphate buffer, and 5 mL of DHAA solution was incubated with EGCG, EGC, ECG, or EC in buffer solution. Four incubation systems were prepared; the total volume was 150 mL. The concentration of DHAA was 2000 µg/mL. The concentrations of EGCG, EGG, ECG, and EC were 458 µg/mL, 442 µg/mL, 306 µg/mL, and 290 µg/mL, respectively. The concentrations of four catechins were all 1 mmol/L. The beaker was incubated at 37 °C in a water bath kettle for 60, 120, 180, 240, 300, 360, 420, 480, and 540 min. The beaker was covered with film, preventing water loss, after which 2 mL of the reaction mixture was taken and then filtered through a 0.22-µm filter for UPLC detection.

### 4.3. Detection Method

The quantification of substrates and products was carried out using a UPLC (Waters, MA, USA), which was equipped with a PDA and a QDA detector. A waters BEH C18 column (1.7 um, 2.1 × 100 mm, Waters, MA, USA) and two mobile phases were used for chromatographic separations. Mobile phase A contained 0.1% formic acid in an aqueous solution, and phase B was pure acetonitrile. The cell temperature was maintained at 15 °C and the column temperature was maintained at 30 °C. The flow rate and injection volume were 350 μL/min and 2 μL/min, respectively. The gradient for solvent B was set as follows: 0–3 min, 60% B; 3–6 min, 90%–83% B; 6–12 min, 83%–73% B; 12–15 min, 73%–50% B; 15–15.5 min, 50% B; 15.5–16 min, 50%–90% B; 16–20 min, 90% B.

The monitoring UV wavelength was set at 280 nm, and the scan range for PDA was 200–500 nm. Mass spectrometry detection was conducted on a QDA in negative ion mode using full-scan recording in a mass rang of *m*/*z* 50–950. A capillary voltage of 0.8 KV, sampling cone voltage of 15 V, source temperature of 120 °C, and desolvation temperature of 600 °C were used.

### 4.4. Statistical Analysis

Watanabe Y and Wang R [26,27] found that EGCG degradation followed first-order kinetics. Hence, EGCG concentration can be expressed by first-order kinetics:Ct/C0 = exp(−kt),(2)
where Ct and C0 are the concentration of EGCG at time t and time = 0, respectively, and where k is the rate constant of degradation. Therefore, the rate constant k of EGCG can be obtained from the gradient of a best-fitted straight line if ln(Ct/C0) is plotted against time.

The rate constant of first-order kinetics was calculated to best fit the experiment results by the regression function of Origin 8.6. All kinetic reactions were performed in at least triplicate.

Statistical analysis was conducted using SPSS 19 software. Statistical significance analysis of different groups was determined by one-way ANOVA with LSD test, while *p* values less than 0.05 were set as the statistical significance level.

## Figures and Tables

**Figure 1 molecules-25-04076-f001:**
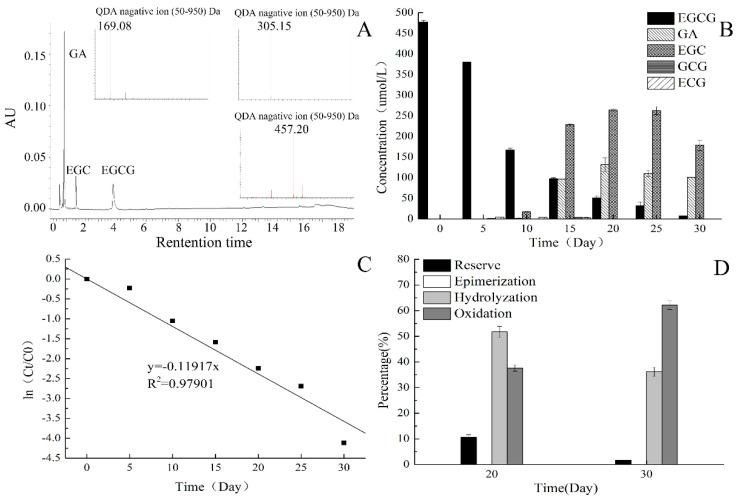
Epigallocatechin gallate (EGCG) stability at 25 °C: (**A**) Chromatograph profile of incubation system stored for 20 days at 25 °C; (**B**) changes in substrates and products during storage of EGCG at 25 °C for 0 to 30 days; (**C**) apparent first-order kinetics of EGCG degradation; (**D**) percentage of the different transformation pathways when EGCG was stored for 20 days and 30 days.

**Figure 2 molecules-25-04076-f002:**
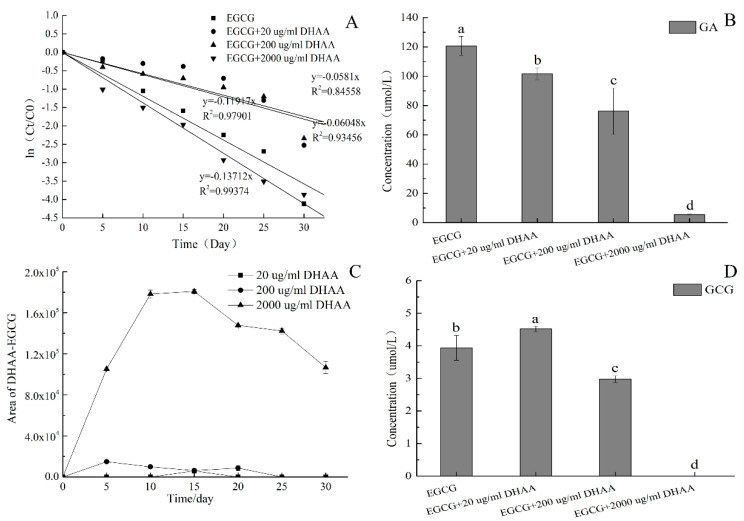
The effect of different dehydroascorbic acid (DHAA) amounts on the EGCG stability: (**A**) Apparent first-order kinetics of EGCG degradation of four incubation systems; (**B**) effect of DHAA on the concentration of GA at the end of reaction (*p* < 0.05); (**C**) changes of DHAA–EGCG area during storage; (**D**) effect of DHAA on the concentration of gallocatechin gallate (GCG) at the end of reaction. Date represent means ± SD of three replicate samples. Different letters indicate significant differences according to Duncan’s multiple range test (*p* < 0.05).

**Figure 3 molecules-25-04076-f003:**
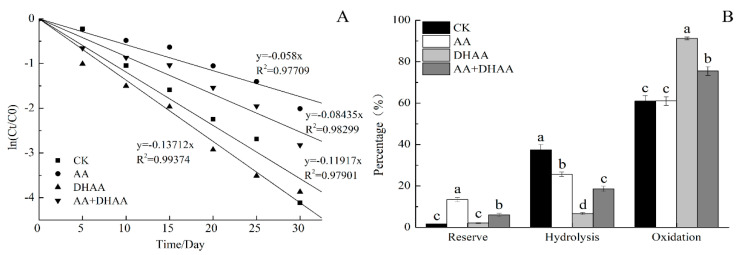
The effect of ascorbic acid (AA) and dehydroascorbic acid (DHAA) on the EGCG stability: (**A**) Apparent first-order kinetics of EGCG degradation of four incubation systems; (**B**) significant analysis of conversion pathway in different incubation systems. Date represent means ±SD of three replicate samples. Different letters indicate significant differences according to Duncan’s multiple range test (*p* < 0.05).

**Figure 4 molecules-25-04076-f004:**
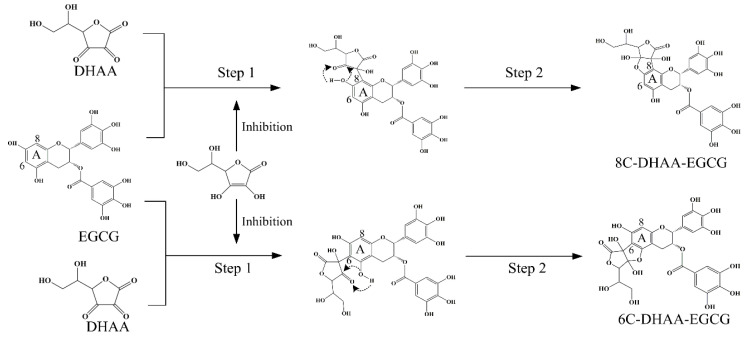
The mechanism for forming DHAA–EGCG in the presence of DHAA with a coexistent AA.

**Figure 5 molecules-25-04076-f005:**
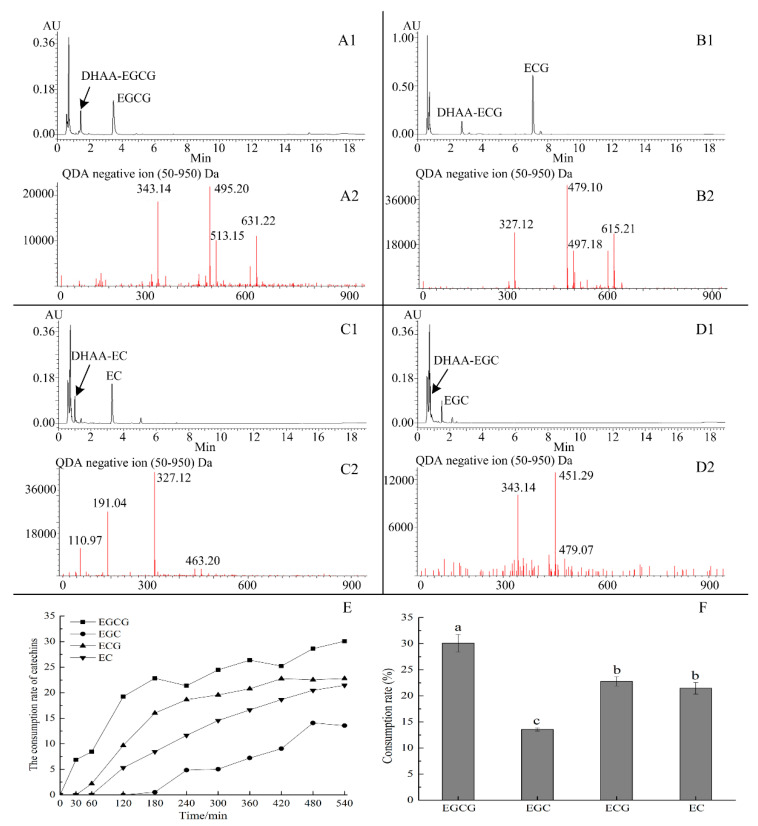
Chromatograph profile and mass spectrum of ascrbyl adducts of catechins. (**A1**,**B1**,**C1**,**D1**) represent the liquid chromatogram of ascorbyl conjunctions of (−)-epigallocatechin gallate (EGCG), (−)-epicatechin (EC), (−)-epicatechin gallate (ECG), and (−)-epigallocatechin (EGC). (**A2**,**B2**,**C2**,**D2**) represent the negative mass spectrum of ascorbyl conjunctions of EGCG, ECG, EC, and EGC, respectively. (**E**) represents the changes of consumption rate of four catechins during storage. (**F**) represents the consumption rate of four catechins at 540 min. Date represent means ± SD of three replicate samples. Different letters indicate significant differences according to Duncan’s multiple range test (*p* < 0.05).

**Table 1 molecules-25-04076-t001:** Precursor ion and characteristic fragment ion of substrates and new products.

Name	Scan Mode	Precursor Ion	Fragment Ion 1	Fragment Ion 2
AA	Negative ion	175.01	114.97	/
DHAA	Negative ion	173.09	110.98	/
EGCG	Negative ion	457.13	305.01	169.07
ECG	Negative ion	305.15	/	/
EC	Negative ion	289.10	/	/
EGC	Negative ion	441.18	/	/
DHAA–EGCG	Negative ion	631.22	495.20	343.14
DHAA–ECG	Negative ion	615.21	479.10	327.12
DHAA–EC	Negative ion	463.20	327.12	191.04
DHAA–EGC	Negative ion	479.07	451.29	343.14

”/” means no detection.

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
