# Peer review of "Dehydroascorbic Acid Affects the Stability of Catechins by Forming Conjunctions"

_molecules, 2020, doi:10.3390/molecules25184076_

Round 1

Reviewer 1 Report

The purpose of the article “Dehydroascorbic acid affects the stability of catechins by forming conjunctions” was to determine the effects of DHAA on the  stability of catechins conducting a series of experiments that incubate DHAA with EGCG or catechins

The article is easy to understand, but I have some requests to make:

1 - Make a list of acronyms and insert at the beginning of the article or as supplementary material

2 - Most of the references used are old (more than 5-10 years). It would be interesting to conduct a review on the subject using more recent articles (main request)

Page 7 – Figure 5: Correct the caption (A3 and A4 is C1 and C2)

Page 9, line 269: What study found that EGCG and tea catechins were stable in pH 5-6?

Reviewer 2 Report

 In general, this study has been very well-organized and English is very clear.  The results are giving an interesting and valuable information to understand the effect of DHAA on EGCG stability.  However, there are some problems and flaws in presentation and discussion.  I hope that my comments are very useful for the improvement of this research.

Major comments

  • Methods: Authors investigated the effects of AA and DHAA on ECGC during storage of green tea. Normally, the headspace of green tea beverage is filled with nitrogen. I think the effect on the change of EGCG depends on the presence or absence of oxygen in the headspace. However, that was not taken into account in this experiment. If the author wishes to discuss the preservation of green tea beverages, authors must consider this point.

Minor comments

  • Fig 1A: The y-axis values span two rows.
  • Fig 1B: In 5 and 10 days, as the storage period increases, EGCG concentrations decrease. But, other compounds (GA, EGC, GCG, ECG) were not increased in 5 and 10 days. If authors know what EGCG is, authors should consider it.
  • Line 262-267: The location of the reagent company (sigma, Shanghai Yuanye biological technology company, and Waters) is not shown.
  • Line 269: This sentence needs to reference of author’s previous study.

Reviewer 3 Report

The manuscript “Dehydroascorbic acid affects the stability of catechins by forming conjunctions” is generally well written and contains data of some relevance for a general readers as well as of high relevance for specialists in the topic. Although the subject of the paper could be of interest for the readers of the journal, the paper needs some corrections:

  • Figure 1:Why were these conditions chosen (20 and 30 days) ?
  • Figure 1 (B) - I suggest changing the symbols/colours for EGCG and ECG
  • Page 3, line 111: There is no explanation of the “CD” symbol
  • Figure 5 – Figure A2 it is not liquid chromatogram. Figures A3 and A4 are missing. In general, check the descriptions of the Figure 5.
  • Page 8, lines from 225 to 259:

In my opinion, the discussion is not exhaustive and should be expanded. Is there more research on this topic? A longer summary is also missing. What are the conclusions of the research? Does the research have any practical application?

  • Page 9, lines from 269 to 270:

Is the research published? Please provide more information on these studies.

Round 2

Reviewer 3 Report

The answers and corrections are satisfactory.